# Profiling of the Candidate Interacting Proteins of SELF-PRUNING 6A (SP6A) in *Solanum tuberosum*

**DOI:** 10.3390/ijms23169126

**Published:** 2022-08-15

**Authors:** Enshuang Wang, Tengfei Liu, Xiaomeng Sun, Shenglin Jing, Tingting Zhou, Tiantian Liu, Botao Song

**Affiliations:** 1Key Laboratory of Horticultural Plant Biology, Ministry of Education, Huazhong Agricultural University, Wuhan 430070, China; 2Key Laboratory of Potato Biology and Biotechnology, Ministry of Agriculture and Rural Affairs, Huazhong Agricultural University, Wuhan 430070, China; 3Potato Engineering and Technology Research Center of Hubei Province, Huazhong Agricultural University, Wuhan 430070, China; 4College of Horticulture and Forestry Science, Huazhong Agricultural University, Wuhan 430070, China; 5State Key Laboratory of Crop Biology, Shandong Key Laboratory of Crop Biology, College of Agronomy, Shandong Agricultural University, Tai’an 271018, China

**Keywords:** screening, Y2H library, StSP6A, interacting proteins, potato

## Abstract

SELF-PRUNING 6A (SP6A), a homolog of FLOWERING LOCUS T (FT), has been identified as tuberigen in potato. StSP6A is a mobile signal synthesized in leaves and transmitted to the stolon through phloem, and plays multiple roles in the growth and development of potato. However, the global StSP6A protein interaction network in potato remains poorly understood. In this study, BK-StSP6A was firstly used as the bait to investigate the StSP6A interaction network by screening the yeast two-hybrid (Y2H) library of potato, resulting in the selection of 200 independent positive clones and identification of 77 interacting proteins. Then, the interaction between StSP6A and its interactors was further confirmed by the Y2H and BiFC assays, and three interactors were selected for further expression analysis. Finally, the expression pattern of *Flowering Promoting Factor 1.1* (*StFPF1.1*), *No Flowering in Short Days 1* and *2* (*StNFL1* and *StNFL2*) was studied. The three genes were highly expressed in flowers or flower buds. *StFPF1.1* exhibited an expression pattern similar to that of *StSP6A* at the stolon swelling stages. *StPHYF*-silenced plants showed up-regulated expression of *StFPF1.1* and *StSP6A*, while expression of *StNFL1* and *StNFL2* was down-regulated in the stolon. The identification of these interacting proteins lays a solid foundation for further functional studies of StSP6A.

## 1. Introduction

Potato (*Solanum tuberosum* L.) is one of the most important crops grown worldwide, and produces an underground storage organ rich in starch. The reproductive pathway of potato can be divided into sexual and asexual reproduction, which correspond to flowering and tuberization, respectively. Tubers are reproductive vegetative organs formed with the underground spreading stems called stolons. It has been well documented that tuberization is controlled by multiple pathways, such as temperature, photoperiod, and sugar. SELF-PRUNING 6A (SP6A), a member of the Flowering LOCUS T (FT) family, belonging to the phosphatidylethanolamine-binding proteins (PEBPs), is a mobile signal that plays an important role in tuberization of potato [1].

As tuberigen, *StSP6A* is expressed in leaves under short-day (SD) conditions and then transmitted to the stolon via phloem [1]. Besides regulation of tuberization, StSP6A also participates in the regulation of flowering and branching. The mechanism for tuberization has been relatively better understood than other phenotypes induced by StSP6A. StSP6A has been demonstrated to be involved in multiple pathways of tuberization, including temperature, microRNA, and photoperiod [2,3,4]. Under long-day (LD) conditions, phytochrome B and phytochrome F form a heterodimer that co-stabilizes the StCOL1 protein. The StCOL1 protein binds to the promoter of *StSP5G* and then promotes its transcription, repressing the expression of *StSP6A* [5]. StSP6A, St14-3-3s, and StFDL1 can form a tuberigen activation complex (TAC), which plays a significant role in tuberization [6]. Recently, StABI5-like 1 (StABL1) was found to bind to StSP6A in a 14-3-3 manner, forming an alternative tuberigen activation complex (aTAC) and thereby resulting in early tuberization [7]. In potato, StSP6A interacts with the SUGARS WILL EVENTUALLY BE EXPORTED TRANSPORTER 11 (StSWEET11) to promote the symplastic transport of sucrose to affect the source-sink pathway of tuberization [8]. It has been also reported that StSP6A is involved in branching and flowering [9,10]. A TCP transcription factor, BRANCHED1b (BRC1b), interacts with StSP6A to promote dormancy and repress tuberization [9,11]. Under SD conditions, the development of flower buds is repressed by the tuberization signal StSP6A [10]. However, the molecular mechanism for StSP6A to mediate flowering remains unclear. Although StSP6A acts as a mobile signal, the mechanism of its movements is currently unclear. However, the specific trafficking mechanisms of FT have been studied. Several regulators that showed critical function for FT long-distance movements have been reported. These regulators interact with FT to aid in FT movement [12,13,14]. In addition, StSP6A exhibits no DNA binding ability, but it is found to regulate some putative downstream target genes (*StMADS1* and *StMADS13*) [6,15]. Therefore, it is necessary to study the interaction proteins of StSP6A.

To comprehensively understand the function of *StSP6A*, it is important to identify its interacting proteins. With the continuous improvement of the yeast two-hybrid (Y2H) technology, many studies have used the Y2H library to explore the candidate interactors of some important proteins in plants and animals [16,17,18,19]. However, there have been few reports of such studies in potato. In this study, to fully identify the interacting proteins of StSP6A in potato, we carried out a comprehensive screening of such proteins in potato by the Y2H method, and identified 77 putative interactors for StSP6A. These interactors will contribute to further functional studies of StSP6A in potato.

## 2. Results

### 2.1. Yeast Two-Hybrid Library Construction

In order to identify the interactors of StSP6A, we generated a Y2H library using CloneMiner II cDNA Library Construction Kit according to the manufacturer’s instructions. Considering that *StSP6A* plays multiple roles in potato, the Y2H library was prepared using a mixture of plant samples, including the photoperiod-sensitive material E26 and photoperiod-insensitive materials E108 and E20. The leaves and stolons were harvested after two days of treatment under LD conditions or SD conditions. The RNA samples were mixed to generate cDNA. The purified cDNA was recombined into the pDONR222, which was then electroporated into *E. coli* DH10B to obtain the primary library. Subsequently, the plasmid of the primary library was extracted and recombined to the pGADT7-DEST. To obtain the secondary library, the recombined pGADT7-DEST was transferred to the DH10B. The plasmid DNA of the secondary library was electroporated into the yeast strain Y187. The number of the independent clones was counted to evaluate the yeast transformation efficiency. The number of independent clones in this library was >3.0 × 10^7^ cell/mL (Appendix A). Twenty-four clones are shown in Appendix A. The length of most fragments ranged from 1000 to 2000 bp.

### 2.2. Screening of StSP6A Interactors

To identify the interactors of StSP6A, we screened the yeast two-hybrid library with the BK-StSP6A as the bait. The yeast strain with BK-StSP6A was further used for Y2H library screening by yeast mating. In total, 200 single clones were transferred to the medium lacking tryptophan, histidine, adenine, and leucine for further screening, and finally 126 positive clones were obtained. Plasmid of positive clones was extracted and then sequenced with the T7 primer. The results revealed that there were 77 independent target proteins (Appendix A), mainly including proteases, heat shock proteins, synthases, and transcription factors, which may perform their functions by interacting with StSP6A.

To further understand the characteristics of these interactors, we performed the Kyoto Encyclopedia of Genes and Genomes (KEGG) and Gene Ontology (GO) enrichment analysis of these candidate interacting proteins. The KEGG enrichment analysis generated eight pathways, and the most popular pathways were “Carbon fixation in photosynthetic organisms” and “Metabolism” (Figure 1A). These interactors were enriched in 40 GO terms, which could be mapped to 20, 15, and 5 GO terms on Biological Process (BP), Cellular Component (CC), and Molecular Function (MF), respectively (Figure 1B). The production of precursor metabolites and energy, photosynthesis, and gene expression of BP were remarkably enriched (Figure 1B). These interactors were widely distributed throughout cells localized in nearly all different subcellular components, including the plastid, ribosome, thylakoid, cytosol, and chloroplast. RNA binding, structural molecule activity, nucleic acid binding, organic cyclic compound binding, and heterocyclic compound binding of MF were also enriched (Figure 1B).

### 2.3. Yeast Two-Hybrid and Bimolecular Fluorescence Complementation Analysis of the Interactors of StSP6A In Vitro and In Vivo

To verify the interaction of StSP6A with the interactors, ten proteins were selected for further study. The full-length CDS of the interactors was amplified and cloned into the pGADT7, which was then co-transformed with BK-StSP6A into the yeast strain AH109. As a result, all of the clones grew normally on the medium lacking tryptophan and leucine, and there were five proteins surviving on the medium without tryptophan, histidine, adenine, and leucine/X-α-Gal (Appendix A). Three proteins related to flowering in *Arabidopsis* were selected for further study. The Y2H and BiFC were conducted to further confirm the interaction between StSP6A and the three interactors. These interactors co-transformed into the yeast with StSP6A showed blue signals in quadruple dropout media (Figure 2A). To further clarify the interaction between StSP6A and the three proteins, the full-length CDS of the three proteins and StSP6A was fused with the split YFP to generate Yc-interactors and Yn-StSP6A, respectively. It could be found that StSP6A interacted with Flowering Promoting Factor (StFPF1.1, Soltu.DM.01G021950.1) in the nucleus and cytoplasm (Figure 2B). No Flowering in Short Days 1 and 2 (StNFL1 and StNFL2, Soltu.DM.02G026970.1 and Soltu.DM.02G007350.1, respectively) interacted with StSP6A in the nucleus (Figure 2B). These results further validated the results of the Y2H library screening.

### 2.4. Subcellular Localization of the StFPF1.1, StNFL1, and StNFL2

To further clarify the subcellular localization of StFPF1.1, StNFL1, and StNFL2, the full-length CDS of these genes was cloned to the C-terminal of green fluorescent protein (GFP). The GFP-StFPF1.1, GFP-StNFL1, GFP-StNFL2, and GFP alone were transiently expressed in *N. benthamiana* leaves by agroinfiltration. As shown in Figure 3, the GFP fusion proteins GFP-StFPF1.1 and GFP-StNFL2 were expressed in the nucleus and cytosol, and GFP-StNFL1 was only expressed in the nucleus (Figure 3). As a transcription factor, StNFL1 might function in regulating some transcriptional events.

### 2.5. Expression Patterns of StFPF1.1, StNFL1, StNFL2, and StSP6A in Response to Tuberization or Flowering

In order to determine whether these interactors are associated with tuberization or flowering, we analyzed their expression patterns in different tissues, distinct stages of stolon swelling, and the *StPHYF*-silenced plants showing tuberization under LD conditions [5]. *StSP6A* showed relatively high expression in tubers. *StFPF1.1* had the highest expression in roots, followed by flowers and then flower buds. The *StNFL1* and *StNFL2* exhibited the highest expression in flower buds. *StNFL1* was highly expressed in flowers, while *StNFL2* was expressed in shoots but not the flowers (Figure 4A), indicating that they might perform some different functions. We further quantified the expression levels of *StSP6A* and its interactors in six distinct stages of the stolon swelling process. Notably, *StFPF1.1* showed an expression pattern similar to that of *StSP6A*, both of which were highly expressed from stage 2 to stage 6. In contrast to *StSP6A*, *StNFL1* showed higher transcription at stage 1 and lower transcription from stage 2 to stage 6. Unlike *StNFL1*, *StNFL2* had a sudden decrease in expression at stage 5 (Figure 4B). *StPHYF* is a photoreceptor playing a critical role in tuberization. We then quantified the transcription levels of *StSP6A* and its interactors in leaves and stolons of *StPHYF*-silenced plants. As expected, *StSP6A* was up-regulated in both leaves and stolons. *StFPF1.1* was hardly expressed in leaves but up-regulated in stolons. The transcription of *StNFL1* and *StNFL2* was suppressed in the *StPHYF*-silenced stolons (Figure 4C), which may act as a repressor of tuberization in potato. These results indicated that *StFPF1.1*, *StNFL1*, and *StNFL2* may participate in the regulation of flowering or tuberization in potato.

## 3. Discussion

*StSP6A* has been identified as a key regulator of tuberization and also participates in the regulation of flowering and branching. However, the molecular mechanism of flowering and branching remains elusive. Some proteins interacting with StSP6A have been reported, including StFDL1, StABL1, StSWEET11, and BRC1b [6,7,8,9,11]. It is important to explore whether there are some other unknown StSP6A interactors involved in other functions. In this study, we performed the Y2H library screening of StSP6A (Appendix A) and identified 77 independent proteins (Appendix A). GO enrichment analysis indicated that the encoded candidate proteins were widely distributed in different subcellular compartments and participated in many critical biological processes (Figure 1B). As StSP6A is localized in the nucleus and cytoplasm, these proteins may perform their functions by interacting with StSP6A (Appendix A). Taken together, these candidate proteins may contribute to a better understanding of the molecular mechanism for the role of *StSP6A* in potato. We then selected three interactors to be verified by Y2H and BiFC assays (Figure 2).

In potato, StSP6A represses the flower bud development under SD conditions [10]. Tuberization shares a conserved pathway with flowering and is regulated by photoperiod. We identified three flowering-related proteins, StNFL1, StNFL2, and StFPF1.1, interacting with StSP6A. The *StNFL1* and *StNFL2* are bHLH family transcription factors, which play an essential role in plant growth and development [21]. In *Arabidopsis*, the *nfl* mutant shows a non-flowering phenotype under SD conditions, which can be rescued by the exogenous application of gibberellin (GA), indicating that NFL acts upstream of GA to promote flowering [22,23]. In this study, two NFLs were identified that interacted with StSP6A in the nucleus (Figure 2B). Both NFL-coding genes were specifically expressed in flowers or flower buds of potato (Figure 4). *StGA2ox1* is a tuberization marker gene, which is upregulated in the *StSP6A*-overexpressing plants [1,24]. There are four binding elements of bHLH transcription factors in the promoter of *StGA2ox1* (Appendix A). Thus, StSP6A may regulate flowering and tuberization by interacting with NFLs and regulating *StGA2ox1*.

*AtFPF1* is an important gene regulating flowering time in *Arabidopsis*. Overexpression of *AtFPF1* in *Arabidopsis* and rice can promote their flowering and initiate the adventitious roots in rice [25,26]. The early flowering induced by *AtFPF1* can be reduced by paclobutrazol and crosses with a GA-deficient mutant, indicating that *AtFPF1* is involved in a GA-dependent signaling pathway and modulates GA response [27]. *OsRAA1*, the homologue gene of *AtFPF1*, showed multiple functions in flower, leaf, and root development in rice, indicating that *FPF1* shares conserved functions in flower and root development [28]. *StFPF1.1* exhibits high expression in roots and flowers (Figure 4). Whether *StFPF1.1* participates in regulating root development and flowering in potato needs to be further studied. *StFPF1.1* is up-regulated in the *StSP6A*-overexpressing plants and *StBEL5*-ox lines [1,29]. In potato, one of the upstream regulators of *StSP6A* is StBEL5, which produces more and longer lateral roots in the overexpression lines [29,30]. Interestingly, *StSP6A* and *StGA2ox1* are up-regulated in stolons and roots of *StBEL5*-ox lines. *StFPF1.1* and *StSP6A* exhibited similar expression patterns during the stolon swelling process and were up-regulated in the stolon of *StPHYF*-silenced plants (Figure 4B,C). In summary, these candidate proteins provide a basis for further studying the potential functions of *StSP6A* in potato.

## 4. Materials and Methods

### 4.1. Plant Materials and Growth Conditions

The potato varieties E20, E108, and E26 were used for the yeast library construction. E20 and E108 are photoperiod-insensitive lines, and E26 is a photoperiod-sensitive line that does not form tubers in LD conditions. Plants were cultured in vitro using single stem nodes in MS medium [31] supplemented with 3% sucrose at 20 °C under LD (light cycle with 16 h:8 h light:dark) conditions, with the light intensity ranging from 400 to 1000 μmol m^−2^s^−1^. Three-week-old plantlets were transplanted into plastic pots with a diameter of 10 cm (one plant per pot) in the growth room under the LD photoperiod of 20 ± 2 °C for 4 weeks, with light intensity of approximately 250 μmol m^−2^s^−1^. The plants were transmitted to the LD or SD (light cycle with 8 h:16 h light:dark) conditions, respectively. After growing for 2 days under LD or SD conditions, the leaves and stolons were harvested at 3–4 h after light on.

E109, which does not form tubers in non-inductive conditions, was used for gene expression experiments. Except for flowers and flower buds, the tissue samples were harvested after growing 4 weeks in LD conditions in vivo. The flower and flower bud were harvested after growing 8 weeks in LD conditions in vivo. All of the samples were harvested at 3 h after light on. The stolons were harvested after 7 days of SD treatment. The leaves and stolons of *StPHYF*-silenced plants were obtained from a previous study [5]. All of the samples were immediately frozen in liquid nitrogen and stored at −70 °C until use.

### 4.2. RNA Extraction and Quantitative Reverse Transcription PCR Analysis

Total RNA was extracted according to the manufacturer’s protocol (Plant Total RNA Kit; ZP405-1; ZOMANBIO, Beijing, China). Monitoring the degradation and contamination of the total RNA samples was by a 1% agarose gel electrophoresis. Reverse transcription into cDNA was performed using 5 × All-In-One RT MasterMix (ABM, Richmond, BC, Canada).

Quantitative reverse transcription PCR was performed with Roche LightCycler 480 Instrument. The program was set according to BrightGreen 2X qPCR MasterMix-No Dye (ABM, Richmond, BC, Canada). The potato *ef1α* gene (GenBank accession: AB061263) was used as an internal reference gene [32]. Gene expression levels were calculated via the 2^−ΔCq^ method (http://www.bio-rad.com/zh-cn/applications-technologies/real-time-pcrexperimental-design (accessed on 4 June 2022). The primer sequence of qRT-PCR is presented in Appendix A.

### 4.3. Yeast Library Construction

The cDNA of E20, E26, and E108 was collected according to the manufacturer’s instructions for the CloneMiner II cDNA Library Construction Kit (Invitrogen, Waltham, MA, USA). The cDNA was recombined with pDONR222 and electroporated into DH10B to construct the primary library. The plasmid extracted from the primary library was recombined into pGADT7-DEST and electrotransferred to DH10B to construct a secondary library [33]. The plasmid of the secondary library was extracted. About 5 μg of plasmid DNA was transformed to the yeast strain Y187, and the transformation efficiency was tested by culturing in SD medium lacking leucine (Clontech, Mountain View, CA, USA) [34,35].

### 4.4. BK-StSP6A Vector Construction

To construct BK-StSP6A, the full-length coding sequence (CDS) of *StSP6A* (Soltu.DM.05G026370.1) was cloned using the Phanta Max Super-Fidelity DNA Polymerase (Vazyme, Nanjing, China) with the gene-specific primers (Appendix A). The CDS of *StSP6A* was recombined into pGBKT7 with *Eco*RI and *Sal*I (Takara, Kusatsu, Japan) sites using Exnase II (Vazyme, Nanjing, China).

### 4.5. Y2H Library Screening and Y2H Assays

The Y2H library screening and Y2H assays were performed according to the protocol as described by the BD Matchmaker Screening Kit (Clontech, Mountain View, CA, USA). The yeast strain AH109 was used in this study. The sequences obtained using T7 primer were mapped to the improved genome assembly and annotation (version 6.1) for the doubled monoploid potato DM 1-3 516 R44 (http://solanaceae.plantbiology.msu.edu/dm_v6_1_download.shtml (accessed on 24 February 2022).

For the Y2H assay, the CDS of the StSP6A candidate interactors was cloned using the gene-specific primers (Appendix A). The CDS of the interactors was recombined into pGADT7 with *Eco*RI and *Bam*HI (Takara, Kusatsu, Japan) sites using Exnase II. The BK-StSP6A and AD-interactors were co-transformed to the AH109 growing on the non-selective double drop-out (SD/-Leu/-Trp, Clontech, Mountain View, CA, USA) for 3 days at 30 °C. Five single clones were selected and grew on the double drop-out and quadruple drop-out (SD/-Leu/-Trp/-His/-Ade, Clontech, Mountain View, CA, USA) media. To the quadruple drop-out medium, 20 mg/L X-α-galactose (X-α-gal) was added in parallel to detect the activation of the α-galactosidase by the protein–protein interactions.

### 4.6. Gene Ontology (GO) and Kyoto Encyclopedia of Genes and Genomes (KEGG) Enrichment Analysis and Visualization

The GO and KEGG enrichment analysis was performed using TBtools with the adjustment *p* < 0.05 [36]. The GO enrichment map was generated by https://www.bioinformatics.com.cn (accessed on 2 June 2022), a free online platform visualization. The KEGG enrichment map was conducted by using the Funny Enrichment Bar Plot of TBtools [36].

### 4.7. Bimolecular Fluorescence Complementation (BiFC) Assays

The full-length *StSP6A* and the CDS of its candidate interactors were generated through amplification using the specific primers (Appendix A) and followed by cloning into pSPYNE-35S and pSPYCE-35S vectors digested with the enzymes *BamH*I and *Sal*I (Takara, Kusatsu, Japan) and recombination using Exnase II [37]. The recombinant plasmid DNA was transformed into the *A. tumefaciens* strain GV3101 by electroporation. The harboring fusion constructs were co-infiltrated into the leaves of *Nicotiana benthamiana* plants at the six-leaf stage. YFP fluorescence was detected after 48 h incubation by a Leica TCS SP8.

### 4.8. Subcellular Localization

For subcellular localization of StNFL1, StNFL2, and StFPF1.1, the full-length coding sequences were cloned into pK7GWF2 at the *Bsp1407*I (Takara, Kusatsu, Japan) sites using Exnase II [38]. GFP fluorescence was detected after 48 h of incubation by a Leica TCS SP8.

## Figures and Tables

**Figure 1 ijms-23-09126-f001:**
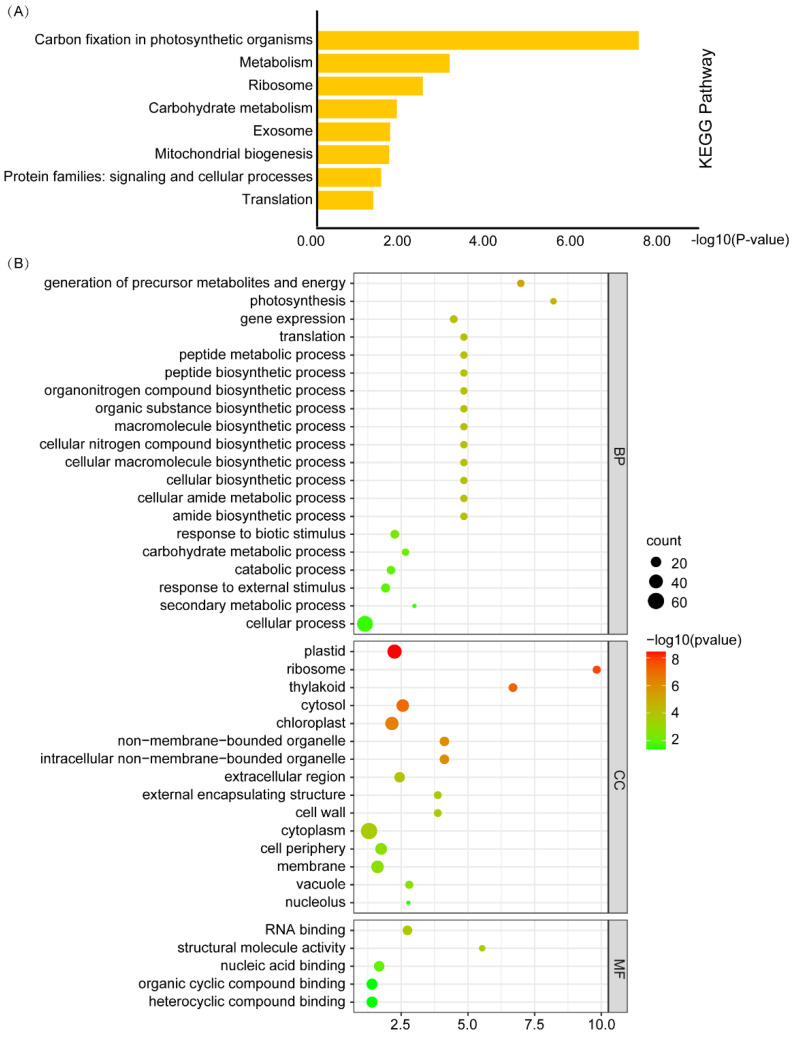
KEGG pathway and GO enrichment analysis of the candidate interactors. (**A**) KEGG pathway enrichment of the StSP6A interactors. (**B**) GO enrichment of the candidate interactors. Each term is represented by a circle node whose size is proportional to the number of input genes falling into the term, and the color represents the −log10 (*p*-value).

**Figure 2 ijms-23-09126-f002:**
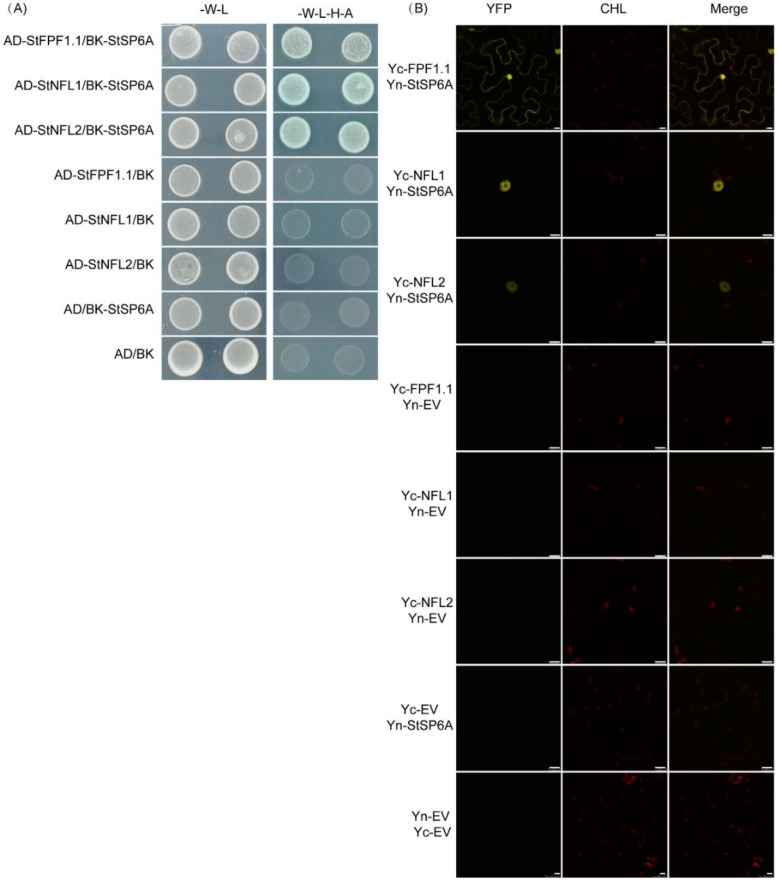
The interactions between the StSP6A and interactors in vivo and in vitro. (**A**) Yeast two-hybrid assays showing the interaction between StSP6A and the interactors. BK-StSP6A and pGADT7 (AD)-interactors were co-expressed in yeast on the double dropout (-Leu-Trp) medium and quadruple dropout medium with 20 mg/L X-α-gal. Empty vectors were used as negative controls. (**B**) BiFC assays to confirm the interaction between StSP6A and the interactors. The *N. benthamiana* leaves were co-transfected with agrobacterium carrying the expression constructs for Yn-StSP6A and Yc-interactors. After 48 h, the YFP fluorescence signals were captured by a Leica TCS SP8 confocal microscope. The bar = 10 μm.

**Figure 3 ijms-23-09126-f003:**
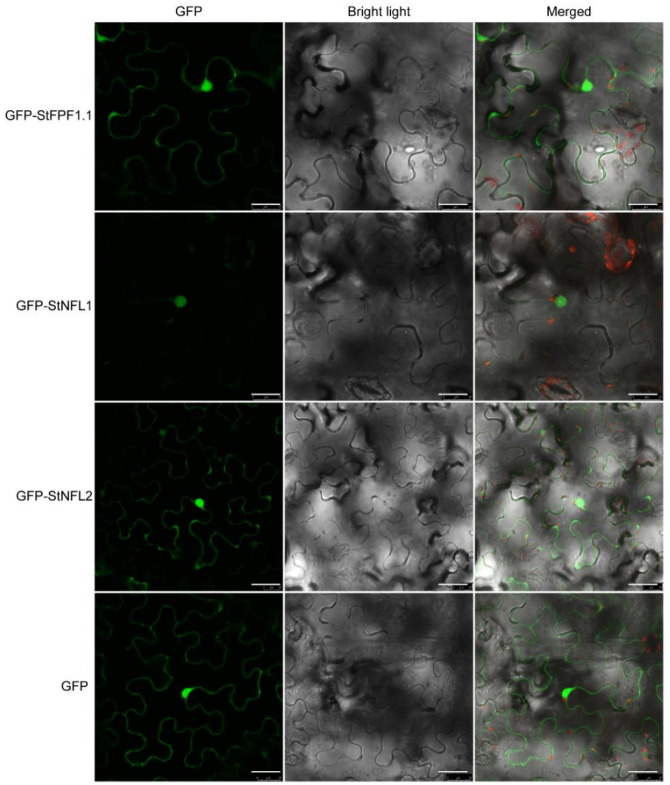
Subcellular localization analysis of the StSP6A interactors. The GFP alone and fusion proteins were transiently expressed in *N. benthamiana* leaves via agrobacterium. After 48 h, the GFP fluorescence signals were captured by a Leica TCS SP8 confocal microscope. GFP alone is localized in the cytoplasm and nucleus. The bar = 25 μm.

**Figure 4 ijms-23-09126-f004:**
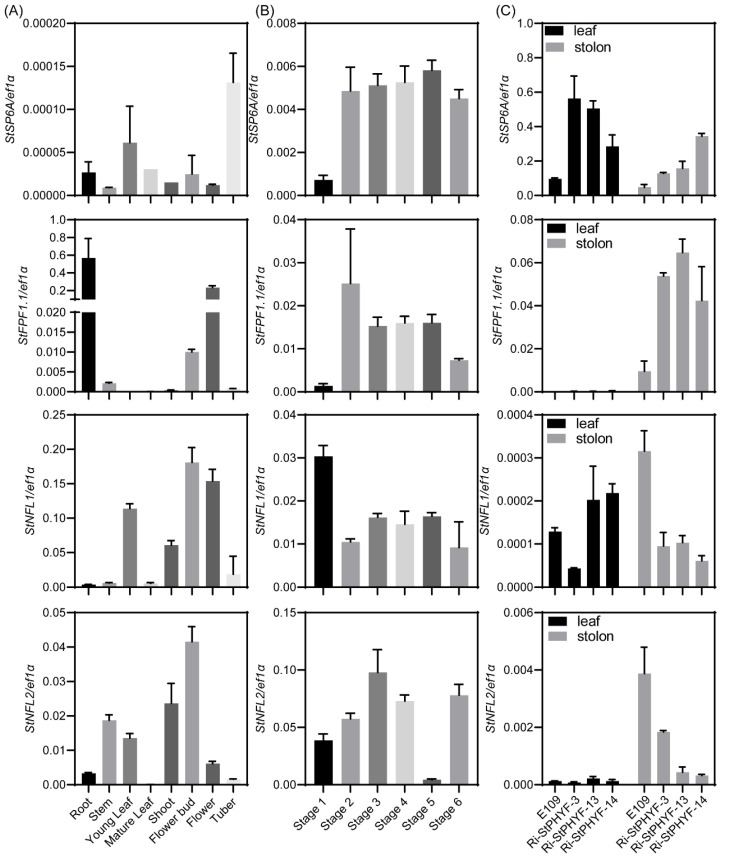
Expression of *StAP6A* and StSP6A interactors in E109 and *StPHYF*-interference plants grown in vitro. (**A**) Relative expression of *StSP6A* and its interactors in different tissues of E109. (**B**) Expression of *StSP6A* and its interactors in stolons of different stages as defined by Begum et al. [20]. (**C**) Expression of *StSP6A* and its interactors in leaves and stolons of *StPHYF*-interference plants with tuberization under non-inductive conditions.

## Data Availability

All data supporting the findings of this research are available within the paper and within its Appendix A published online.

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
