# Peer review of "Profiling of the Candidate Interacting Proteins of SELF-PRUNING 6A (SP6A) in Solanum tuberosum"

_ijms, 2022, doi:10.3390/ijms23169126_

Round 1
Reviewer 1 Report
Dear authors, the paper you intend to publish in IJMS mdpi journal has been read and revized. The document is very good and I have a minor revisions.
1.- Please use the ID of potato genes within the document and also in supplementary information. . It is very difficult to read, example: Soltu.DM. 06G027300.2.
2.- The ID of flowering promoting factor is PGSC0003DMT400029505, you can use the name StFPF1.1 (PGSC0003DMT400029505), in order to better understand the manuscript. ID of NFL1 in potato is LFY.
Line 193 correct the extra parenthesis
((A)Relative expression of StSP6A and its interactors in different tissues of E109. 193 (B) Expression of StSP6A and its interactors in stolons of different stages shown with Shahnewaz 194
Line 194 use the name of Begum instead of Shahnewaz
Shahnewaz 194 [22]. (C)
Discussion.
You wrote: As StSP6A is a key regulator of tuberization, the mechanism of tuberization is relatively clear.
You mean that, it has been described that StSP6A is a key regulator of tuberization, but the actual molecular mechanism of tuber development is not known. What is relatively clear? If it is not known.
Line 231, complete adequately this:
And StBEL5 acts upstream of StSP6A [30]. 231
BEL5 regulates StSP6 and in the reference 30 (Sharma 2016) they found many genes (At least 10 000) activated.
Explain:
What kind of protein is StSP6A?: Explain why did you get apart of flowering genes ribosomal proteins.
References:
Reference 11 is not well written
Tang, D.; Jia, Y.; Zhang, J.; Li, H.; Cheng, L.; Wang, P.; Bao, Z.; Liu, Z.; Feng, S.; Zhu, X.; Li, D.; Zhu, G.; Wang, H.; Zhou, Y.; 341 Zhou, Y.; Bryan, G. J.; Buell, C. R.; Zhang, C.; Huang, S., Genome evolution and diversity of wild and cultivated potatoes. Nature 342 2022.
Cite as: Tang D, Jia Y, Zhang J, Li H, Cheng L, Wang P, Bao Z, Liu Z, Feng S, Zhu X, Li D, Zhu G, Wang H, Zhou Y, Zhou Y, Bryan GJ, Buell CR, Zhang C, Huang S. Genome evolution and diversity of wild and cultivated potatoes. Nature. 2022 Jun;606(7914):535-541. doi: 10.1038/s41586-022-04822-x.
Reference 27: is not well written
Guo, Y.; Wu, Q.; Xie, Z.; Yu, B.; Zeng, R.; Min, Q.; Huang, J., OsFPFL4 is involved in the root and flower development by 376 affecting auxin levels and ROS accumulation in rice (Oryza sativa). Rice 2020, 13, (1), 1-15.
Cite as: Guo, Y., Wu, Q., Xie, Z. et al. OsFPFL4 is Involved in the Root and Flower Development by Affecting Auxin Levels and ROS Accumulation in Rice (Oryza sativa). Rice 13, 2 (2020). https://doi.org/10.1186/s12284-019-0364-0
Author Response
Response to Reviewer 1 Comments
Dear authors, the paper you intend to publish in IJMS mdpi journal has been read and revized. The document is very good and I have a minor revision.
Point 1: Please use the ID of potato genes within the document and also in supplementary information. It is very difficult to read, example: Soltu.DM. 06G027300.2.
Response 1: Thank you for your significant comments. The sequences obtained by the T7 promoter were mapped to the improved genome assembly and annotation (version 6.1) for the doubled monoploid potato DM 1-3 516 R44. The improved potato genes ID were used in the document and supplementary information. This information was added to the materials and methods part (page 9, line 279-281).
Point 2: The ID of flowering promoting factor is PGSC0003DMT400029505, you can use the name StFPF1.1 (PGSC0003DMT400029505), in order to better understand the manuscript. ID of NFL1 in potato is LFY.
Response 2: Thanks for your valuable advice. The ID of flowering promoting factor with version 6.1 is Soltu.DM.01G021950.1. We change the name of StFPF1.1, StNFL1 and StNFL2 to StFPF1.1 (Soltu.DM.01G021950.1), StNFL1 (Soltu.DM.02G026970.1), and StNFL2 (Soltu.DM.02G007350.1), respectively. In potato, the ID of floral meristem identify control protein LEAFY (LFY) is Soltu.DM.03G032420.1. Comparing the CDs of LFY and NFL1, it showed 32.73% identity(Fig.1).
Figure 1. Nucleotides alignment of StLFY and StNFL1 CDs.
Point 3: Line 193 correct the extra parenthesis
((A)Relative expression of StSP6A and its interactors in different tissues of E109. 193 (B) Expression of StSP6A and its interactors in stolons of different stages shown with Shahnewaz 194. Line 194 use the name of Begum instead of Shahnewaz
Shahnewaz 194 [22]. (C)
Response 3: Thanks for your helpful suggestions. The errors were corrected as suggested.
Discussion.
Point 4: You wrote: As StSP6A is a key regulator of tuberization, the mechanism of tuberization is relatively clear.
You mean that, it has been described that StSP6A is a key regulator of tuberization, but the actual molecular mechanism of tuber development is not known. What is relatively clear? If it is not known.
Response 4: Sorry for the confusion. Now it reads The mechanism of tuberization is clear since StSP6A has been identified as a key regulator of tuberization. StSP6A also participates in the regulation of flowering and branching, but the molecular mechanism remains elusive
Point 5: Line 231, complete adequately this:
And StBEL5 acts upstream of StSP6A [30]. 231
BEL5 regulates StSP6 and in the reference 30 (Sharma 2016) they found many genes (At least 10 000) activated.
Explain:
Response 5: Thank you for your helpful comments. StBEL5, as a mobile TF, that interacts with a partner protein regulates a complex network of cellular growth, expansion, and quiescence. In Sharma et al research, StBEL5 functions as an activator in cytokinin and auxin pathways, and a repressor in the GA pathway of growth. Both StSP6A and StFPF1.1 are upregulated in the StBEL5-overexpression lines. StBEL5 appears to be positioned upstream of an intricate regulatory network involving hormonal pathways and transcriptional regulators that controls the onset of tuber formation.
Point 6: What kind of protein is StSP6A?: Explain why did you get apart of flowering genes ribosomal proteins.
Response 6: Thanks for your kind comments. StSP6A, the homolog of FT, belongs to the phosphatidylethanolamine-binding proteins (PEBPs). The PEBP family protein plays multiple roles in plant growth and development. The StSP6A is localized in the cytoplasm and nucleus. The ribosomal proteins were identified in this study. The abundance of ribosomal protein is high and can be easily identified. It has been reported that the ribosomal proteins are patriated in regulating flowering (Ito et al 2000; Degenhardt et al 2008; Li et al 2022). Whether these identified ribosomal proteins are involved in the regulation of flowering or tuberization needs further study.
References:
Reference 11 is not well written
Tang, D.; Jia, Y.; Zhang, J.; Li, H.; Cheng, L.; Wang, P.; Bao, Z.; Liu, Z.; Feng, S.; Zhu, X.; Li, D.; Zhu, G.; Wang, H.; Zhou, Y.; 341 Zhou, Y.; Bryan, G. J.; Buell, C. R.; Zhang, C.; Huang, S., Genome evolution and diversity of wild and cultivated potatoes. Nature 342 2022.
Cite as:Tang D, Jia Y, Zhang J, Li H, Cheng L, Wang P, Bao Z, Liu Z, Feng S, Zhu X, Li D, Zhu G, Wang H, Zhou Y, Zhou Y, Bryan GJ, Buell CR, Zhang C, Huang S. Genome evolution and diversity of wild and cultivated potatoes. Nature. 2022 Jun;606(7914):535-541. doi: 10.1038/s41586-022-04822-x.
Response: Corrected as suggested.
Reference 27: is not well written
Guo, Y.; Wu, Q.; Xie, Z.; Yu, B.; Zeng, R.; Min, Q.; Huang, J., OsFPFL4 is involved in the root and flower development by 376 affecting auxin levels and ROS accumulation in rice (Oryza sativa). Rice 2020, 13, (1), 1-15.
Cite as: Guo, Y., Wu, Q., Xie, Z.et al. OsFPFL4 is Involved in the Root and Flower Development by Affecting Auxin Levels and ROS Accumulation in Rice (Oryza sativa). Rice 13, 2 (2020). https://doi.org/10.1186/s12284-019-0364-0
Response: Corrected as suggested.

Reviewer 2 Report
The authors have done a fundamental work and revealed the new insights of function and interaction of SP6A in potato.
In my opinion, the manuscript can be published in IJMS.
I just have a minor comment:
- Gene names as well as scientific names should be provided in italic format, please check entire text.
Author Response
Response to Reviewer 2 Comments
The authors have done a fundamental work and revealed the new insights of function and interaction of SP6A in potato.
In my opinion, the manuscript can be published in IJMS.
I just have a minor comment:
Gene names as well as scientific names should be provided in italic format, please check entire text.
Response: Thanks for your kind suggestion. The gene names were corrected to the italic format.
